# A Treatment Paradigm Shift: Targeted Radionuclide Therapies for Metastatic Castrate Resistant Prostate Cancer

**DOI:** 10.3390/cancers14174276

**Published:** 2022-09-01

**Authors:** Ephraim E. Parent, Adam M. Kase

**Affiliations:** 1Department of Radiology, Mayo Clinic, 4500 San Pablo Road, Jacksonville, FL 32224, USA; 2Division of Hematology Oncology, Mayo Clinic, Jacksonville, FL 32224, USA

**Keywords:** prostate cancer, radionuclide therapy, bone, PSMA, GRPR, somatostatin

## Abstract

**Simple Summary:**

Metastatic prostate cancer has traditionally been treated with a combination of hormonal and chemotherapy regimens. With the recent FDA approval of targeted radionuclide therapeutics, there is now a new class of therapy that is routinely available to patients and clinicians. This review explores the most commonly studied therapeutic radiopharmaceuticals and their appropriate use and contraindications. Additionally, we detail how these therapeutic radiopharmaceuticals can fit into the common medical oncology practice and future directions of this field of medicine.

**Abstract:**

The recent approval of ^177^Lu PSMA-617 (Pluvicto^®^) by the United States Food and Drug Administration (FDA) is the culmination of decades of work in advancing the field of targeted radionuclide therapy for metastatic prostate cancer. ^177^Lu PSMA-617, along with the bone specific radiotherapeutic agent, ^223^RaCl_2_ (Xofigo^®^), are now commonly used in routine clinical care as a tertiary line of therapy for men with metastatic castrate resistant prostate cancer and for osseus metastatic disease respectively. While these radiopharmaceuticals are changing how metastatic prostate cancer is classified and treated, there is relatively little guidance to the practitioner and patient as to how best utilize these therapies, especially in conjunction with other more well-established regimens including hormonal, immunologic, and chemotherapeutic agents. This review article will go into detail about the mechanism and effectiveness of these radiopharmaceuticals and less well-known classes of targeted radionuclide radiopharmaceuticals including alpha emitting prostate specific membrane antigen (PSMA)-, gastrin-releasing peptide receptor (GRPR)-, and somatostatin targeted radionuclide therapeutics. Additionally, a thorough discussion of the clinical approach of these agents is included and required futures studies.

## 1. Introduction

Prostate cancer (PCa) is the second most common cancer among men in the United States, with one out of eight men diagnosed during their lifetime [1]. When identified early, patients with PCa can undergo highly curative therapy with definitive radical prostatectomy or radiotherapy. However, up to 30% of patients with PCa will eventually develop metastatic castration-resistant prostate cancer (mCRPC), as prostate cancer becomes androgen independent [2,3]. Despite androgen independence, androgen deprivation therapy remains the backbone of treatment, in addition to, bone modifying agents and cancer-directed therapy. Metastatic disease to the bone poses great morbidity with skeletal-related events and pain, overall, negatively impacting quality of life. Bone modifying agents such as bisphosphonates (zoledronic acid) and receptor activator of nuclear factor κ B ligand (RANKL) inhibitor (denosumab) are necessary in CRPC patients with bone metastases to prevent SREs which are known to increase the risk of death and reduce quality of life [4,5]. There are multiple cancer directed therapeutic options available that improve overall survival (OS) in mCRPC which include androgen signaling inhibitors (abiraterone, enzalutamide), chemotherapy (docetaxel, cabazitaxel), autologous cellular immunotherapy (sipuleucel-T) and poly-ADP-ribose polymerase inhibitors (olaparib, rucaparib); however, despite these systemic therapies, mCRPC remains incurable [6]. Advances in the field of targeted radionuclide therapy for mCRPC has led to the widespread adoption of bone specific radionuclide therapy (^223^Ra dichloride; Xofigo^®^) and prostate-specific membrane antigen (PSMA) targeted radiotherapy (^177^Lu PSMA-617; Pluvicto^®^) (Table 1). In this review, we will discuss these United States Food and Drug Administration (FDA) approved radiotherapeutics for mCRPC and discuss other radionuclide therapies in development including alpha (α) emitting PSMA radiopharmaceuticals, gastrin-releasing peptide receptor (GRPC) targeted α/β emitting radiopharmaceuticals, and somatostatin targeted radionuclide therapy (^177^Lu DOTATATE, Lutathera^®^).

## 2. Bone Specific Radiotherapeutics

Nearly 90% of patients with mCRPC will ultimately develop osseous metastatic disease leading to pain and negatively impacting quality of life [14]. There have been several alpha-(α) and beta-(β) emitting bone specific therapeutic radiopharmaceuticals for men with mCRPC in development over the years with ^223^Radium dichloride (^223^RaCl_2_; Xofigo^®^) becoming the first FDA approved agent in prostate cancer in 2013. Compared to other radiopharmaceutical agents analogous to ^223^RaCl_2_, ^223^RaCl_2_ has an advantage due to its short half-life of 11.4 days and decay predominately through α-emission, allowing for high linear energy transfer (LET) and high amounts of double-stranded DNA breaks when in the decay pathway. Other previously utilized bone targeted radionuclides (phosphorus-32, samarium-153, strontium-89) decayed through β-emission which results in a lower LET and fewer DNA breaks [15]. ^223^RaCl_2_ physiologically behaves like calcium and forms complexes with bone matrix hydroxyapatite, preferentially being incorporated into areas of high bone turnover which is typically seen in osteoblastic bone metastases, the predominant form of osseous disease in patients with mCRPC [15]. ^223^RaCl_2_ is rapidly cleared from the blood with only 20% of the injected dose remaining in the blood 15 min after injection, and at 4 h 61% is localized to the skeleton with the remaining 39% in the bowel for subsequent fecal elimination. Given the fecal route of elimination, dose adjustments for patients with hepatic or renal dysfunction are not necessary. ^223^RaCl_2_ is administered intravenously at 55 KBq/kg every 4 weeks for 6 cycles. As ^223^RaCl_2_ decays via α particles, which have a negligible path length in air, patients can be immediately released to go home after administration.

The Alpharadin in Symptomatic Prostate Cancer Patients (ALSYMPCA) trial was a phase III randomized double-blind placebo-controlled trial with 921 patients who had symptomatic mCRPC with two or more bone metastases detected with skeletal scintigraphy and without evidence of visceral metastatic disease. Patients were enrolled to receive either ^223^RaCl_2_ every 4 weeks for 6 cycles or placebo, the study met the primary endpoint of an improved OS of 14.9 mo vs. 11.3 mo (Hazard ratio (HR) 0.70; 95% CI, 0.58 to 0.83; *p* < 0.001) [7]. Secondary endpoints including time to first symptomatic skeletal event (SSE), time to rise in alkaline phosphatase, and prostate specific antigen (PSA) progression were also improved in the ^223^RaCl_2_ arm and there were no significant differences in adverse events between the two groups. Perhaps more importantly, the quality of life was improved in the ^223^RaCl_2_ group based on validated instruments: EuroQol 5-dimentsion 5- level (EQ-5D) and Functional Assessment of Cancer Therapy-Prostate (FACT-P) [16]. A secondary reanalysis also found that patients on the ^223^RaCl_2_ arm also had fewer hospitalization days per patient (4.44 vs. 6.68; *p* = 0.004) in the first year after treatment and improvement in pain compared to the placebo group [17].

The FDA approved ^223^RaCl_2_ for the use in patients with mCRPC who have symptomatic bone metastases and no visceral disease. While this remains the primary indication for treatment with ^223^RaCl_2_, there have been several studies demonstrating that ^223^RaCl_2_ may also benefit men with asymptomatic bone disease. In a single arm prospective study with 708 patients, asymptomatic (n = 135, 19%) patients were more likely to complete therapy with ^223^RaCl_2_ compared to symptomatic (n = 548, 77%) patients; in addition, overall survival (HR 0.486), time to progression (HR 0.722), and time to first SSE (HR 0.328) were better in asymptomatic patients compared to symptomatic patients [18]. There have also been efforts to incorporate the use of ^223^RaCl_2_ in mCRPC patients with visceral metastases, given that most patients with mCRPC have a large component of bony disease regardless of their visceral involvement [14]. Assessing treatment response to ^223^RaCl_2_ with molecular imaging remains a challenge with commonly utilized bone specific radiopharmaceuticals (e.g., ^99m^Tc methylene diphosphonate (MDP) as both benign healing and metastatic disease can have a similar presentation (Figure 1). Increases in PSA levels, which often portend progression of disease, are often seen with ^223^RaCl_2_ treatment and should not be relied upon in the decision to stop ^223^RaCl_2_ [19]. Additionally, while treatment with ^223^RaCl_2_ has been shown to lead to drops in alkaline phosphatase and lactate dehydrogenase levels, these markers are also not dependable to determine the effectiveness of ^223^RaCl_2_ [20]. As ^223^RaCl_2_ localizes to the bone marrow, blood counts should be monitored to ensure the absolute neutrophil count (ANC) is ≥1 × 10^9^/L and platelets are ≥50 × 10^9^/L before each treatment with ^223^RaCl_2_. If hematologic values do not recover 6–8 weeks after the last ^223^RaCl_2_ treatment, ^223^RaCl_2_ should be discontinued.

## 3. Beta Emitting PSMA Targeted Radiotherapeutics

PSMA is a transmembrane glutamate carboxypeptidase that is highly expressed in prostate cancer and has become a leading target in diagnostic imaging and a powerful new therapeutic target. PSMA is expressed in more than 90% of metastatic PCa lesions and demonstrates higher expression with greater Gleason scores [21,22]. Given the differential expression of PSMA between PCa and normal tissues, small molecule PSMA targeted radiotherapeutics have been developed for prostate cancer, such as the FDA approved ^177^Lu PSMA-617 (Pluvicto^®^) and the promising non-FDA approved ^177^Lu PSMA I&T. The benefit of this targeted molecular therapy is based on the binding, internalization, and retention of the PSMA ligands within tumor cells [23].

^177^Lu PSMA-617 was FDA approved on 23 March 2022 for the treatment of patients with PSMA-positive mCRPC and who have been treated with an androgen receptor (AR) pathway inhibitor and taxane-based chemotherapy [24]. PSMA PET is essential to identify patients with mCRPC who will benefit PSMA-targeted radioligand therapy (RLT) [25], with beta (e.g., Lu-177) or alpha (Ac-225) PSMA radiotherapeutics [26,27] (Figure 2). There are currently two FDA approved PSMA PET radiopharmaceuticals for patients with suspected prostate cancer metastasis who are candidates for initial definitive therapy or suspected recurrence based on elevated PSA levels: ^68^Ga PSMA-11 (Ga 68 gozezotide, Illuccix^®^, Locametz^®^) and ^18^F DCFPyL (Pifluofolastat F 18, Pylarify^®^). The FDA package insert for ^177^Lu PSMA-617 (Pluvicto^®^) specifies that patients selected for treatment must use the FDA approved PSMA PET radiopharmaceutical ^68^Ga PSMA-11 (Illuccix^®^, Locametz^®^) to confirm the presence of PSMA-positive disease [24]. However, of note, NCCN guidelines state that PET imaging with either ^68^Ga PSMA-11 or ^18^F DCFPyL can be used to determine eligibility for ^177^Lu PSMA-617 therapy [28]. Additionally, Novartis has announced a strategic collaboration with Lantheus to include ^18^F DCFPyL in clinical trials with Lu PSMA-617 RLT, suggesting ^18^F DCFPyL PET may be acceptable in the future prior to ^177^Lu PSMA-617 RLT [29].

Two major multicenter clinical trials, VISION (USA and Canada) and TheraP (Australia), investigated the outcome of patients with mCRPC after ablation with ^177^Lu PSMA-617 [12,13]. The phase III VISION trial evaluated ^177^Lu PSMA-617 in 831 patients with mCRPC and was the principal justification for FDA approval of ^177^Lu PSMA-617 RLT. Primary outcomes measured radiographic progression-free survival (rPFS) and OS between ^177^Lu PSMA-617 RLT plus SOC versus standard of care (SOC) alone. When compared to SOC alone, ^177^Lu PSMA-617 plus SOC significantly prolonged rPFS (median, 8.7 vs. 3.4 months; HR for progression or death 0.40; 99.2% CI, 0.29 to 0.57) and median OS (15.3 vs. 11.3 months; HR for death, 0.62; 95% CI, 0.52 to 0.74; *p* < 0.001). The phase II TheraP trial, compared ^177^Lu PSMA-617 to cabazitaxel in 200 men with mCRPC. The primary endpoint was PSA response defined by a reduction of PSA ≥ 50% from baseline. In contrast to the VISION trial, TheraP set PSMA SUVmax requirements of at least one lesion on ^68^Ga-PSMA-11 PET with SUV_max_ > 20, and the remaining metastatic lesions SUV_max_ > 10, and no discordant hypermetabolic disease. PSA responses were more frequent among men in the ^177^Lu PSMA-617 group versus the cabazitaxel group (66% vs. 37%, respectively).

The TheraP trial outcomes are considered superior to the VISION trial, likely as the result of exclusion of mCRPC patients with discordant hypermetabolic lesions. While the VISION trial used conventional imaging to exclude patients with discordant lesions (positive lesions on CT and negative on PSMA PET), the TheraP trial used functional techniques including ^18^F-fludeoxyglucose (FDG) PET/CT in conjunction with PSMA PET/CT, and patients with at least one discordant hypermetabolic lesion, PSMA (−)/FDG (+), were excluded. Patients with mCRPC and with discordant hypermetabolic lesions have been shown to have worse outcomes and discordant hypermetabolic disease is often seen in a sizable minority of patients with mCRPC [30,31]. In a study of 56 patients, Chen et al. found that 23.2% had at least one PSMA (−)/FDG (+) lesion, and that PSA and Gleason score were both higher in these patients with discordant hypermetabolic disease [32]. A sub-analysis of a single center phase II trial of ^177^Lu PSMA-617 RLT similarly found that 16/50 patients had at least one PSMA (−)/FDG (+) lesion and were deemed ineligible for ^177^Lu PSMA-617 therapy. The OS of these patients with discordant hypermetabolic disease was 2.6 months (compared to 13.5 months for patients that received ^177^Lu PSMA-617) [33].

While the FDA package insert for ^177^Lu PSMA-617 does not specify any contraindications to therapy, the EANM guidelines have published contraindications for PSMA-RLT [26]. For the most part, these guidelines have mirrored the inclusions and exclusion criteria of large phase II/III trials such as VISION [12] and TheraP [13] with some minor variations. These contraindications include: (1) Life expectancy is less than 6 months and ECOG performance status > 2. (2) Unacceptable medical or radiation safety risk. (3) Unmanageable urinary tract obstruction or hydronephrosis. (4) Inadequate organ function (GFR < 30 mL/min or creatinine > 2-fold upper limit of normal (ULN); liver enzymes > 5-fold ULN). (5) Inadequate marrow function (with total white cell count less than 2.5 × 10^9^/L or platelet count less than 75 × 10^9^/L). (6) Conditions (e.g., spinal cord compression and unstable fractures) which require timely interventions (e.g., radiation therapy and surgery) and in which PSMA-RLT might be performed afterwards depending upon the patient’s condition.

General radiation safety precautions should be followed with ^177^Lu-PSMA RLT, with local and national guidelines dictating specific clinical practice. Radiation safety precautions may be modeled after ^177^Lu-DOTATATE therapy for neuroendocrine tumors given a shared radionuclide [26,34]. A recent meta-analysis of ^177^Lu PSMA-617 dosimetry found that the lacrimal and salivary glands are the critical organs with the kidneys also receiving a significant radiation dose [35]. The calculated radiation absorbed doses to the lacrimal and salivary glands after 4 cycles of ^177^Lu PSMA-617 is near the tolerated dose limit whereas the dose to the kidneys is far below the dose tolerance limits. ^177^Lu PSMA-617 has been shown to have a low, but significant, rate of adverse events (AE) in several clinical studies. In the phase III VISION study, 52.7% of patients experienced a grade 3 or higher AE, as compared to 38.0% of patients with similar events in the control group. Anemia was the most common grade ≥3 AE, observed in 12.9% of subjects. Additionally, a recently published meta-analysis of 250 studies with a total of 1192 patients similarly found that while grade 3 and 4 toxicities were uncommon, anemia was the highest reported adverse event for both ^177^Lu PSMA-617 (0.19 [0.06–0.15]) and ^177^Lu PSMA—I&T (0.09 [0.05–0.16]) [36]. Greater than 35% of patients in the treatment group of the VISION trial experienced fatigue, dry mouth, or nausea, though almost entirely grade ≤ 2 AE [12]. Adverse event incidence was similar to smaller early phase studies that preceded the VISION study [13,37,38,39].

## 4. Dosimetry and Future Developments of PSMA Targeted Radiotherapeutics

Utilizing dosimetry to tailor dosing to a patient’s particular biology has potential to potentiate the benefits of ^177^Lu PSMA-617 RLT. While the large TheraP [13] and VISION [12] trials employed a fixed dosing of 200 mCi (7.4 GBq), a small study demonstrated safety of dosing of up to 250 mCi (9.3 GBq) in selected cohorts [40]. In principle, a patient-centered dosing scheme can calculate a safe maximum tolerated activity and maximize radiation dose to tumors [41,42]. This need to augment ^177^Lu PSMA-617 dosage is underscored by a study that showed that patients receiving less than 10 Gy to tumors were unlikely to achieve a PSA response (≥50% PSA decline in pretreatment PSA) [43]. Additionally, recent studies have demonstrated a “tumor sink” effect, where patients with particularly high burden disease demonstrated reduced delivery of ^68^Ga-PSMA-11 [44] or ^177^Lu PSMA-617 [45] to target tissues. Unfortunately, the ability of the treating physician to prescribe a tailored dose of ^177^Lu PSMA-617 to patients is currently almost non-existent in the United States, given the one-size-fits-all approach Novartis has employed of providing a fixed dose of 200 mCi per cycle of ^177^Lu PSMA-617.

There are several open questions and innovations that promise to further extend the role of ^177^Lu PSMA-617 in PCa. For example, the synergistic effects from combination therapies as well as the appropriate sequencing of the treatment in the disease course remain uncertain. Both VISION and TheraP were deployed late in mCRPC disease when patients have limited therapeutic options remaining. Both trials demonstrate ^177^Lu-PSMA-617 RLT to be effective at improving clinical outcomes; however, patients may also benefit if therapy is employed earlier in their disease course. Several trials are currently underway in hopes of answering this question. The UpFrontPSMA and PSMAddition trials seek to determine the efficacy and safety of ^177^Lu PSMA-617 in men with metastatic hormone-sensitive prostate cancer. Other trials are assessing ^177^Lu PSMA-617 as first-line therapy for mCRPC or in the neoadjuvant setting for localized PCa.

## 5. Alpha Emitting PSMA Targeted Radiotherapeutics

Another area of emerging interest is the use of α emitting radioisotopes for PSMA targeted radiotherapy. Actinium-225 is an α emitting radioisotope that has been chelated to several PSMA chemical ligands, including PSMA-617 [46]. Kratochwil et al. [47] reported two patients who had complete responses to ^225^Ac PSMA-617, including one who had previously progressed after ^177^Lu PSMA-617 treatment. This initial report has been confirmed in several small case series [46,48]. Pooling 10 small studies together, a recent meta-analysis found a 62.8% PSA50 (decrease in PSA ≥50% compared to baseline) response rate for ^225^Ac PSMA-617 [49]. Particular attention to evaluating ^225^Ac PSMA-617 in mCRPC patients that have failed previous lines of therapy, including ^177^Lu PSMA-617, is ongoing. The high LET and different microdosimetry in tumors exposed to α particles is seen to overcome cellular defences when resistance to β emitters (e.g., Lu-177) is found [50,51]. A retrospective analysis of 26 men with progressive mCRPC that had undergone several previous therapies, including ^177^Lu PSMA-617, found that ^225^Ac PSMA-617 resulted in a ≥50% PSA drop in 65% of patients [52], but with greater hemotoxicity and permanent xerostomia [46] than in patients with less advanced disease [53]. Of note, the short path length of α particles is especially valuable in the treatment of patients with extensive skeletal metastatic disease, with the goal of protecting the normal bone marrow from the AE seen with ^177^Lu PSMA-617 as previously discussed [54]. In a retrospective study of patients treated with ^225^Ac PSMA-617, 106 patients were found to have either multifocal (≥20) skeletal metastases (n = 72, 67.9%), or a diffuse pattern of axial skeletal involvement with or without appendicular skeletal involvement (i.e., superscan pattern) on ^68^Ga PSMA-11. Eighty-five of the 106 patients (80.2%) treated with ^225^Ac PSMA-617 achieved a PSA response of ≥50% and had only rare hematologic toxicity with renal dysfunction being a significant risk factor [55]. As ^225^Ac/ ^177^Lu-PSMA radiopharmaceuticals have different benefits and risks, small trials have also incorporated a “tandem” therapy strategy with small doses of ^225^Ac-PSMA being administered together with ^177^Lu-PSMA and with promising results [56]. One major challenge in the clinical use of ^225^Ac-PSMA beyond the scope of small research studies is the limited availability of the isotope itself, but there are many ongoing efforts to increase the global supply of ^225^Ac and other α-emitting radioisotopes.

## 6. Gastrin-Releasing Peptide Receptor (GRPR) Targeted Radiotherapeutics

While efforts towards clinical applications of gastrin-releasing peptide receptor (GRPR) targeted radionuclide therapy are behind those of PSMA targeted radiopharmaceuticals, GRPR is a prime target for radionuclide therapy in men with mCRPC who may have failed β/α PSMA therapy. GRPR (also known as bombesin receptor 2 (BB2)) is a transmembrane receptor expressed on the surface of many cancers and is overexpressed in most PCa [57,58]. Bombesin is a 14-amino acid peptide agonist that binds with high affinity to GRPR and has been shown to increase the motility and metastatic potential of prostate cancer cells [59]. Many diagnostic and therapeutic radiopharmaceuticals have been developed using bombesin as the pharmaceutical core for targeted diagnostic and radiotherapeutic pairs for PCa [60,61,62,63,64]. The bombesin agonist ^177^Lu AMBA demonstrated potential therapeutic effectiveness in several preclinical prostate cancer tumor models [65], but a phase I dose escalation study in patients with mCRPC was stopped due to severe adverse effects due to GRPR stimulation at the therapeutic levels of administered ^177^Lu AMBA [66] and most other GRPR targeted radiotherapeutic agonists have encountered similar safety problems. Conversely, GRPR antagonists do not appear to cause any adverse side effects and most recent efforts have concentrated on GRPR antagonists. The GRPR antagonist ^177^Lu RM2 has been evaluated in mCRPC patients with high uptake in prostate cancer cells and demonstrates rapid clearance from physiologic GRPR expressing tissues, such as the pancreas [67]. An additional highly potent GRPR antagonist, NeoBOMB1, is being evaluated in a multicenter study as a combined diagnostic/therapeutic drug with ^68^Ga/^177^Lu, respectively, [68]. One major problem with the bombesin-derived diagnostic and therapeutic radiopharmaceuticals is the rapid proteolytic degradation due to peptidases [69,70] with several biochemical modifications being explored in bombesin analogs including unnatural residues and peptidase inhibitors. As with PSMA, GRPR expression is modified by several hormonal and immunomodulators and the effectiveness of ^177^Lu RM2 was found to be potentiated with the addition of the mTOR inhibitor rapamycin in preclinical trials [71]. Combination GRPR targeted radionuclide therapy and immunotherapy with ^177^Lu RM26 and trastuzumab, respectively, lead to the synergistic therapy of prostate cancer in mice models [72].

## 7. Somatostatin Targeted Radiotherapeutics

Although a de novo clinical presentation of small cell neuroendocrine carcinoma of the prostate is rare, a subset of patients previously diagnosed with prostate adenocarcinoma may develop neuroendocrine features in later stages of mCRPC progression [73]. Neuroendocrine prostate cancer (NePC) is an aggressive variant of prostate cancer that most frequently retains early PCa genomic alterations and acquires new molecular changes making them resistant to traditional mCRPC therapies and AR targeted therapies have little effect [74]. Some of the difficulty in treating patients with mCRPC may be due to neuroendocrine differentiation [75]. Of particular importance, NePC is notorious for having little to no PSMA expression, resulting in no appreciable role for either PSMA PET imaging or ^177^Lu/^225^Ac PSMA targeted radionuclide therapy. Somatostatin, a neuropeptide that suppresses prostate growth and neovascularization by inducing cell-cycle arrest and apoptosis, is highly expressed in NePC cells (Figure 3) [76,77]. Somatostatin receptors have also been shown to be upregulated in prostate adenocarcinoma [78,79]. Preliminary case reports suggest that ^68^Ga-DOTA labeled somatostatin analogs may have high sensitivity in identifying sites of mCRPC in addition to NePC [80,81,82,83]. In a recent study involving 12 patients with mCRPC, all patients had at least 1 blastic neuroendocrine metastasis with increased ^68^Ga-DOTA uptake [84]. The large degree of somatostatin expression in NePC and mCRPC, suggests that ^177^Lu-DOTATATE (Lutathera) may be an alternative to β/α PSMA therapy PSMA, either if having failed PSMA targeted radiotherapy or in the cases with no or little PSMA expression on PSMA PET. While ^177^Lu-DOTATATE is used extensively for neuroendocrine carcinoma, there are only a couple of case reports of patients with NePC that have been treated with ^177^Lu-DOTATATE with initial success [85]. This area requires further attention to demonstrate if it is a viable target for directed radionuclide therapy.

## 8. Discussion and Clinician’s Perspective

Understanding how to incorporate the two FDA approved radiopharmaceutical therapies, ^223^RaCl_2_ (Xofigo^®^) and ^177^Lu-PSMA-617 (Pluvicto^®^) into the treatment paradigm of mCRPC is essential to maximize their therapeutic potential. While all FDA approved agents for mCRPC offer an absolute overall survival benefit, compared to their control arm, this incremental benefit only approaches 5 months for each therapy. Therefore, allowing the patients the opportunity to receive as many therapies as possible is paramount to derive the maximum survival benefit. The optimal sequence of these therapies is lacking, either in the literature or routine clinical practice; however, when selecting treatment, the clinician should consider the disease burden, tempo of disease, location of metastases, prior therapies utilized and anticipated therapies. The chosen sequence often depends on the provider’s philosophy on treatment which could be aimed at aggressive approaches upfront to achieve timely disease control while the patient is fit enough to receive therapy, or a clinician may meet the tempo of disease with therapies that offer control of the disease with the least toxicity. These aspects of cancer care delivery should be considered when incorporating ^223^RaCl_2_ or ^177^Lu-PSMA-617. The FDA approved label for ^223^RaCl_2_ allows for the treatment of symptomatic mCRPC patients with bone metastases, detected with conventional skeletal scintigraphy, without evidence of visceral metastatic disease. While all patients were symptomatic in the ALSYMPCA trial, symptomatic pain was broadly defined and opioid pain control was not required, and 44% of patients had only mild pain with nonopioid therapy at baseline and these patients also achieved a survival benefit when compared to placebo [86]. Therefore, ^223^RaCl_2_ should be considered earlier in the disease course when quality of life is still preserved. To further investigate the efficacy of ^223^RaCl_2_ surrounding chemotherapy, a prespecified subgroup analysis showed survival benefit was maintained regardless of prior docetaxel use [87]. This survival benefit is important to note because many patients that could benefit from ^223^RaCl_2_ are not candidates for chemotherapy or may decline chemotherapy. It is reported that 20–40% of patients with CRPC may not receive chemotherapy [7]; therefore, this targeted radionuclide therapy remains a possibility for patients who have not been exposed to chemotherapy, especially as the current indication for ^177^Lu-PSMA-617 requires previous chemotherapy exposure. With triple therapy on the horizon (i.e., chemotherapy plus androgen receptor pathway inhibitor and ADT), understanding that a benefit can be achieved with ^223^RaCl_2_ after chemotherapy remains applicable to future patient populations who might receive chemotherapy in the metastatic hormone-sensitive setting. To further optimize ^223^RaCl_2_ efficacy, combination therapy is being investigated. In the ERA-223 trial, abiraterone acetate/prednisone was combined with ^223^RaCl_2_; however, the trial was unblinded early after more fractures and deaths were observed within the combination group [8]. The use of bone protective agents (BPA) was low in this cohort at 40% which led to the mandatory incorporation of BPA in the ongoing phase III EORTC-1333-GUGG trial (PEACE III trial) with enzalutamide plus ^223^RaCl_2_. The phase III trial is being investigated since the phase II trial with enzalutamide plus ^223^RaCl_2_ met its primary endpoint of decreasing bone metabolic markers and was associated with improved outcomes [9]. While not sufficiently powered to determine a true significant difference, the phase II secondary endpoints of OS, rPFS, and time to next treatment were longer in the combination group, 30.8 months vs. 20.6 months (*p* = 0.73), 11.5 months vs. 7.35 months (*p* = 0.96), and 15.9 months vs. 3.47 months (*p* = 0.067), respectively [10]. This suggests a potential role of combination therapy with ^223^RaCl_2_ plus enzalutamide which will be further determine based on the PEACE III results. As ^223^RaCl_2_ is designed for bone predominant disease, combination therapy with enzalutamide would allow for incorporating ^223^RaCl_2_ in patients who have both bone and lymph node disease to potentially acquire the survival benefits that both therapies offer. In a phase II open-label single arm study, as part of an expanded access program analysis, ^223^RaCl_2_ was determined to be safe regardless of concurrent androgen signaling inhibitor. In addition, ^223^RaCl_2_ survival was longer for patients who received less than 3 anticancer therapies [11]. In conclusion, ^223^RaCl_2_ remains a therapeutic option for symptomatic bone predominant disease with or without previous docetaxel exposure and should be incorporated earlier in the sequence of therapy to achieve the largest benefit. In addition, the clinician could consider concurrent therapy with enzalutamide to not only target bone disease, but also non-osseous lesions.

^177^Lu-PSMA-617 is FDA approved for mCRPC patients previously treated with an androgen receptor pathway inhibitor (ARPI) and taxane-based chemotherapy. This radiopharmaceutical therapy is dependent on the presence of PSMA-positive lesions seen on ^68^Ga-PSMA-11 PET imaging [88]. With the approval of ^177^Lu-PSMA-617, clinicians now have a low toxicity therapeutic option for heavily pre-treated patients. In contrast to ^223^RaCl_2_ which does not result in radiologic responses, in patients with measurable or non-measurable disease at baseline who received ^177^Lu-PSMA-617, the objective response rate (ORR) was 29.8% (vs. 1.7% control arm) and a complete response (CR) was achieved in 18 patients (5.6%). These complete responses are remarkable given ^177^Lu-PSMA-617 was given in at least the third-line setting. These responses were based on RECIST v1.1 [12,13] and OS, with radiologic progression or response based on CT, MRI, or bone scintigraphy. Additionally, disease control was achieved in 89.0% of patients. Therefore, clinicians could consider assessing treatment response based on conventional imaging, rather than with PSMA PET as this imaging modality could be cost prohibitive.

In regard to PSA response, 46% of patients in the ^177^Lu-PSMA-617 arm had a PSA response of ≥50% compared to the SOC alone arm of only 7.1% [88]. In the VISION trial, SOC predominately included gonadotropin-releasing hormone analogues, ARPI, bone protective agents and glucocorticoids. In the ^177^Lu-PSMA-617 arm, 54.7% of patients received concurrent abiraterone or enzalutamide as part of their standard of care and 77.5% of patients in the standard of care alone arm received abiraterone or enzalutamide. A survival subgroup analysis was performed on patients based on presence of concurrent ARPI therapy with ^177^Lu-PSMA-617. In patients who received ^177^Lu-PSMA-617 plus ARPI the hazard ratio for death was 0.55 (95%, 0.43–0.70) and patients who received ^177^Lu-PSMA-617 without ARPI the hazard ratio for death was 0.70 (0.53–0.93) [89]. Therefore, survival benefit was achieved regardless of the addition of an ARPI; however, uncertainty remains whether concurrent therapy could increase the efficacy further.

Clinical and pre-clinical studies have shown that ARPI, such as enzalutamide, can enhance PSMA expression with possible potentiation of the effect of ^177^Lu-PSMA-617 therapy [90,91]. Further studies (ENZA-p) are ongoing to investigate the added benefit of concurrent therapy (^177^Lu-PSMA-617 plus enzalutamide vs. enzalutamide alone), therefore, clinicians should consider financial toxicity and added adverse effects when considering concurrent RLT plus ARPI vs. RLT alone. The next consideration is how to incorporate ^177^Lu-PSMA-617 into the current treatment sequence. Prior to ^177^Lu-PSMA-617 FDA approval, cabazitaxel was established as the next therapeutic option after progressing on an ARPI and docetaxel based on the CARD trial. In this trial, cabazitaxel resulted in a mOS of 13.6 months vs. 11.0 months (HR 0.64; 95% CI, 0.46 to 0.89; *p* = 0.008) in patients treated with an ARPI not previously used (abiraterone or enzalutamide) [92]. The TheraP trial investigated the activity and safety of cabazitaxel vs. ^177^Lu-PSMA-617 in patients with metastatic CRPC and who received prior docetaxel treatment. The ^177^Lu-PSMA-617 treatment group achieved higher PSA responses compared to cabazitaxel, 66% vs. 37% (*p* ≤ 0.0001), respectively and had less grade 3–4 adverse events, 33% vs. 53%. After a median follow-up of 3 years, there was no survival difference between groups (19.1 months vs. 19.6 months; restricted mean survival team of 3 years). Since survival appears to be similar between these two agents it is important to contrast again the eligibility criteria used in the VISION and TheraP trial. In the VISION trial patients were required to have ≥ one PSMA-positive lesion and no PSMA-negative soft tissue-or visceral lesions ≥ 1 cm or PSMA-negative lymph nodes ≥ 2.5 cm; in the TheraP trial, patients underwent both FDG-PET and PSMA-PET imaging and patients were excluded if there were discordance (PSMA negative/FDG positive) since these patients have a poor survival with a median OS of 2.5 months [33]. While the majority of patients will meet these criteria for exclusively PSMA avid disease, clinicians should be aware of these conditions and consider chemotherapy with cabazitaxel or platinum-based chemotherapy if patients have significant visceral disease or non-PSMA lesions as outlined above and consider ^177^Lu-PSMA-617 plus ARPI if no other treatment strategies are available.

Since ^177^Lu-PSMA-617 is approved after ARPI and taxane-based therapy and without other contraindication, ^177^Lu-PSMA-617 has the potential to be used after ^223^RaCl_2_ posing concern for myelotoxicity. As mentioned earlier adequate bone marrow function is a prerequisite for treatment with ^177^Lu-PSMA-617, therefore, it was hypothesized that previous chemotherapy or radiation (i.e., ^223^RaCl_2_) could impact candidacy for ^177^Lu-PSMA-617 RLT. A retrospective study was performed of 28 patients who received ^177^Lu-PSMA-617 within 8- weeks after the last ^223^RaCl_2_ administration. Grade ≥ 3 hematologic toxicity was seen in 6 patients with anemia (17.9%), leukopenia (14.3%), and thrombocytopenia (21.4%) which appears similar to hematologic toxicity seen in the VISION trial. Regardless, adequate bone marrow function at the start of ^177^Lu-PSMA-617 is necessary. Given the overall survival benefit and complete radiographic responses seen with ^177^Lu-PSMA-617, this therapeutic option should be prioritized after progression on ARPI and taxane-based therapy. However, as discussed in this review, there is still much work that needs to be accomplished to evaluate these radionuclide therapeutics in earlier stages of disease and there are multiple ongoing clinical trials investigating the role of targeted radionuclide therapeutics for prostate cancer in conjunction with hormonal, chemotherapeutic, and immunologic treatments as stand along therapies (Table 2).

## 9. Conclusions

With the introduction of multiple radiopharmaceuticals into clinical practice, there is a shift in the treatment paradigm for patients with advanced prostate cancer and clinicians are faced with determining how best to sequence these therapies. Given the success of ^223^RaCl_2_ and ^177^Lu PSMA-617, targeted radionuclide therapeutics are now seen as a viable and important adjunct to the therapeutic algorithm that clinicians utilize. Several other classes of promising targeted radionuclide radiopharmaceuticals, both alpha and beta emitters, are also being explored and posed to complement existing treatment algorithms for prostate cancer.

## Figures and Tables

**Figure 1 cancers-14-04276-f001:**
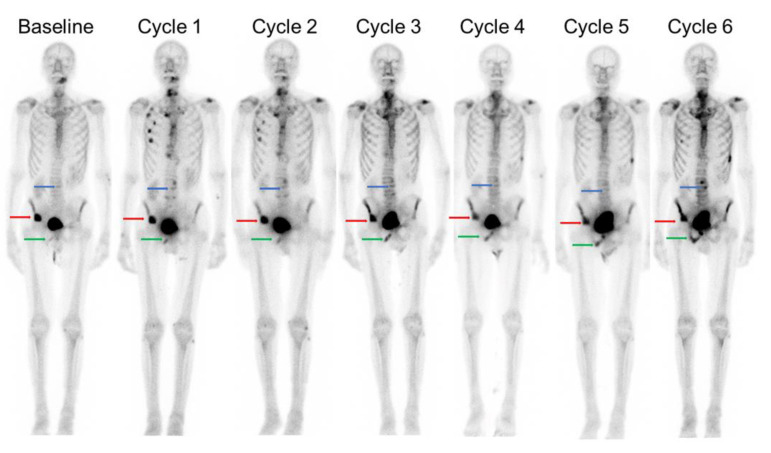
^99m^Tc MDP bone scintigraphy evaluation of ^223^RaCl_2_ therapy. Gleason score 4 + 3 = 7 prostate cancer undergoing serial ^99m^Tc methylene diphosphonate (MDP) bone scans and with known osseous metastatic deposits (arrows) during treatment with 6 cycles of ^223^RaCl. Patient is concurrently maintained on Lupron, and bone protective therapy with abiraterone and prednisone. At baseline prior to ^223^RaCl_2_, he was treated with oxycodone for pain control and had a baseline PSA of 10.9 ng/mL. Lack of quantitative measurements limits the standard planar evaluation of response to therapy. Serial MDP bone scintigraphy demonstrated some improvement in the right iliac metastatic deposit (red arrow) with ^223^RaCl therapy but progressive disease in the right inferior pubic ramus (green arrow) and lumbar spine (blue arrow). Note right sided post traumatic rib fractures at cycle 1 and cycle 2.

**Figure 2 cancers-14-04276-f002:**
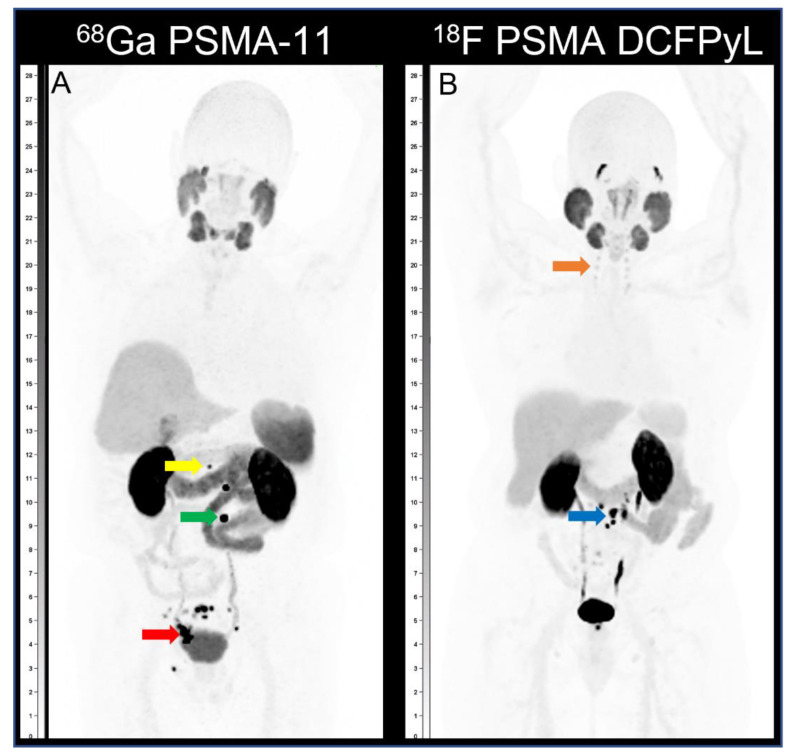
^68^Ga PSMA-11 and ^18^F PSMA DCFPyL PET evaluation of mCRPC prior to ^177^Lu PMSA-617 RLT. Two patients with mCRPC undergoing PSMA PET prior to ^177^Lu PMSA-617 therapy. Patient A with Gleason score 3 + 4 = 7 prostate cancer status post prostatectomy, salvage radiation- and cryotherapy. PSA of 0.42 ng/mL at time of ^68^Ga PSMA-11 PET/CT for evaluation prior to ^177^Lu PSMA-617 therapy. Anterior view of ^68^Ga PSMA-11 PET maximum intensity projection (MIP) (**A**) demonstrates intense PSMA uptake along the prostatectomy bed and rectum (red arrow) retroperitoneal and pelvic lymph nodes (green arrow) and osseous metastatic deposit involving the L1 vertebral body (yellow arrow). Patient B with Gleason 4 + 5 = 9 prostate cancer status post radiation therapy and androgen deprivation therapy, and PSA of 3 ng/mL. Anterior view MIP (**B**) demonstrates intense PSMA uptake along retroperitoneal lymph nodes (blue arrow). Incidental note of symmetric PSMA uptake along benign celiac ganglia (orange arrow).

**Figure 3 cancers-14-04276-f003:**
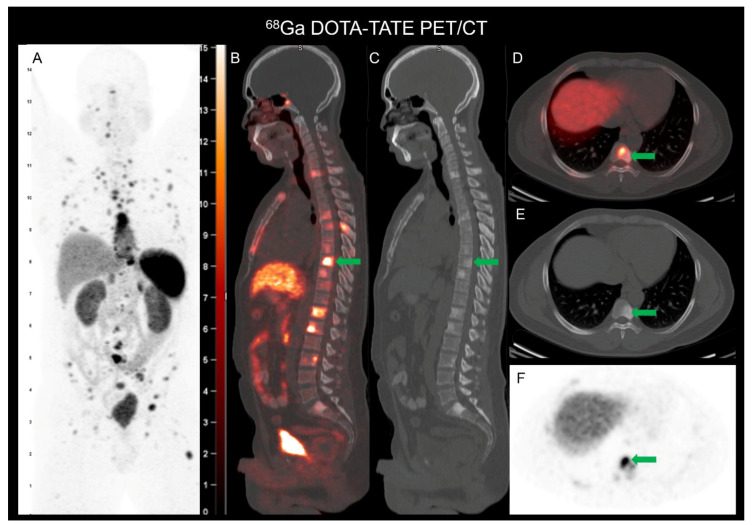
^68^Ga DOTATATE PET/CT evaluation of small cell neuroendocrine prostate carcinoma. Patient with Gleason score 5 + 4 = 9 mixed prostate small cell neuroendocrine carcinoma and acinar adenocarcinoma. Patient was started on ADT and cisplatin/etoposide prior to ^68^Ga DOTATATE PET/CT. Anterior view of ^68^Ga DOTATATE PET MIP (**A**) demonstrates multiple ^68^Ga DOTATATE osseous and nodal metastatic deposits. Selected sagittal fused ^68^Ga DOTATATE PET/CT (**B**) and CT (**C**) images show marked ^68^Ga DOTATATE uptake greater than liver (SUVmax of 14.4) in several osseous lesions. Transaxial fused ^68^Ga DOTATATE PET/CT (**D**) and CT (**E**) and PET (**F**) images show marked ^68^Ga DOTATATE in the most avid T8 lesion having a SUVmax of 20.1 (green arrow). Patient did not demonstrate a PSA response to therapy and passed away 4 months after ^68^Ga DOTATATE PET/CT.

**Table 1 cancers-14-04276-t001:** Pivotal Phase II/III studies leading to FDA approval of ^223^RaCl_2_ and ^177^Lu PSMA-617.

** ^223^ ** **RaCl_2_**
Alpha Emitter Radium-223 and Survival in Metastatic Prostate Cancer (ALSYMPCA) [7]	^223^RaCl_2_ vs. placebo in mCRPC with bone metastasis	Phase III	^223^RaCl_2_ improved overall survival vs. placebo (median, 14.0 months vs. 11.2 months).
Addition of radium-223 to abiraterone acetate and prednisone or prednisolone in patients with castration-resistant prostate cancer and bone metastases (ERA 223) [8]	Abiraterone acetate + prednisone/prednisolone with ^223^RaCl_2_ vs. placebo	Phase III	Addition of ^223^RaCl_2_ did not improve symptomatic skeletal event-free survival and was associated with increasing frequency of fractures (9% vs. 3%).
Prospective Evaluation of Bone Metabolic Markers as Surrogate Markers of Response to Radium-223 Therapy in Metastatic Castration-resistant Prostate Cancer [9,10]	Enzalutamide + ^223^RaCl_2_ vs. enzalutamide alone	Phase II	Combination Enzalutamide + ^223^RaCl_2_ did not show increase in fractures or other adverse events and showed improved bone metabolic markers.
Radium-223 Safety, Efficacy, and Concurrent Use with Abiraterone or Enzalutamide: First U.S. Experience from an Expanded Access Program [11]	^223^RaCl_2_ + concurrent abiraterone acetate or enzalutamide	Phase II	Patients with less advanced disease (<3 prior therapies) were more likely to benefit from ^223^RaCl_2_
** ^177^ ** **Lu PSMA-617**			
Lutetium-177–PSMA-617 for Metastatic Castration-Resistant Prostate Cancer [12]	^177^ Lu PSMA-617 +SOC vs. SOC alone	Phase III	^177^Lu PSMA-617 +SOC (compared to SOC alone) improved rPFS (median, 8.7 vs. 3.4 months) and OS (median, 15.3 vs. 11.3 months).
[^177^Lu]Lu-PSMA-617 versus cabazitaxel in patients with metastatic castration-resistant prostate cancer (TheraP): a randomized, open-label, phase 2 trial [13].	^177^ Lu PSMA-617 vs. cabazitaxel	Phase III	^177^Lu PSMA-617 arm had greater PSA response (65%) vs. cabazitaxel (37%) Grade 3–4 adverse events occurred in (33%) of 98 men in the ^177^Lu PSMA-617 v 45 (53%) of 85 men in the cabazitaxel group.

**Table 2 cancers-14-04276-t002:** Current ongoing targeted radionuclide clinical trials for prostate cancer [91].

ClinicalTrials.gov Identifier	Name of Study	Study Sponsor	Trials Phase	Location
**PSMA**				
NCT04443062	Lutetium-177-PSMA-617 in Oligo-metastatic Hormone Sensitive Prostate Cancer (Bullseye)	Radboud University Medical Center	Phase 2	The Netherlands
NCT05114746	Study of ^177^Lu-PSMA-617 In Metastatic Castrate-Resistant Prostate Cancer in Japan	Novartis Pharmaceuticals	Phase 2	Japan
NCT05079698	A Study of Stereotactic Body Radiotherapy and ^177^Lu-PSMA-617 for the Treatment of Prostate Cancer	Memorial Sloan Kettering Cancer Center	Phase 1	New York, USA
NCT03454750	Radiometabolic Therapy (RMT) With ^177^Lu PSMA 617 in Advanced Castration Resistant Prostate Cancer (CRPC) (LU-PSMA)	Istituto Scientifico Romagnolo per lo Studio e la cura dei Tumori	Phase 2	Italy
NCT05219500	Targeted Alpha Therapy With ^225^Actinium-PSMA-I&T of Castration-resISTant Prostate Cancer (TATCIST)	Excel Diagnostics and Nuclear Oncology Center	Phase 2	Texas, USA
NCT04343885	In Men With Metastatic Prostate Cancer, What is the Safety and Benefit of Lutetium-^177^PSMA Radionuclide Treatment in Addition to Chemotherapy (UpFrontPSMA)	Peter MacCallum Cancer Centre	Phase 2	Australia
NCT04419402	Enzalutamide With Lu PSMA-617 Versus Enzalutamide Alone in Men With Metastatic Castration-resistant Prostate Cancer (ENZA-p)	Australian and New Zealand Urogenital and Prostate Cancer Trials Group	Phase 2	Australia
NCT03780075	^177^Lu-EB-PSMA617 Radionuclide Treatment in Patients With Metastatic Castration-resistant Prostate Cancer	Peking Union Medical College Hospital	Phase 1	China
NCT03874884	^177^Lu-PSMA-617 Therapy and Olaparib in Patients With Metastatic Castration Resistant Prostate Cancer (LuPARP)	Peter MacCallum Cancer Centre	Phase 1	Australia
NCT05162573	EBRT + Lu-PSMA for N1M0 Prostate Cancer (PROQURE-1)	The Netherlands Cancer Institute	Phase 1	The Netherlands
NCT04769817	ProsTIC Registry of Men Treated With PSMA Theranostics	Peter MacCallum Cancer Centre	Observational	Australia
NCT04689828	^177^Lu-PSMA-617 vs. Androgen Receptor-directed Therapy in the Treatment of Progressive Metastatic Castrate Resistant Prostate Cancer (PSMAfore)	Novartis Pharmaceuticals	Phase 3	Multinational
NCT04597411	Study of ^225^Ac-PSMA-617 in Men With PSMA-positive Prostate Cancer	Endocyte	Phase 1	Australia
NCT04886986	^225^Ac-J591 Plus ^177^Lu-PSMA-I&T for mCRPC	Weill Medical College of Cornell University	Phase 1/2	New York, USA
NCT05340374	Cabazitaxel in Combination With ^177^Lu-PSMA-617 in Metastatic Castration-resistant Prostate Cancer (LuCAB)	Peter MacCallum Cancer Centre	Phase 1/2	Australia
NCT05204927	Lu-177-PSMA-I&T for Metastatic Castration-Resistant Prostate Cancer	Curium US LLC	Phase 3	USA
NCT04647526	Study Evaluating mCRPC Treatment Using PSMA [Lu-177]-PNT2002 Therapy After Second-line Hormonal Treatment (SPLASH)	POINT Biopharma	Phase 3	Multinational
NCT04996602	Therapeutic Efficiency and Response to 2.0 GBq (55mCi) ^177^Lu-EB-PSMA in Patients With mCRPC	Peking Union Medical College Hospital	Phase 1	China
NCT04720157	An International Prospective Open-label, Randomized, Phase III Study Comparing ^177^Lu-PSMA-617 in Combination With SOC, Versus SOC Alone, in Adult Male Patients With mHSPC (PSMAddition)	Novartis Pharmaceuticals	Phase 3	Multinational
NCT05113537	Abemaciclib Before ^177^Lu-PSMA-617 for the Treatment of Metastatic Castrate Resistant Prostate Cancer (UPLIFT)	Vadim S Koshkin	Phase 1	California, USA
NCT04946370	Maximizing Responses to Anti-PD1 Immunotherapy With PSMA-targeted Alpha Therapy in mCRPC	Weill Medical College of Cornell University	Phase 1/2	New York, USA
NCT04868604	^64^Cu-SAR-bisPSMA and ^67^Cu-SAR-bisPSMA for Identification and Treatment of PSMA-expressing Metastatic Castrate Resistant Prostate Cancer (SECuRE)	Clarity Pharmaceuticals Ltd.	Phase 1/2	USA
NCT05230251	Radioligand fOr locAl raDiorecurrent proStaTe cancER (ROADSTER)	Glenn Bauman, Lawson Health Research Institute	Phase 2	Canada
NCT04576871	Re-treatment ^225^Ac-J591 for mCRPC	Weill Medical College of Cornell University	Phase 1	New York, USA
NCT04726033	^64^Cu-TLX592 Phase I Safety, PK, Biodistribution and Dosimetry Study (CUPID Study) (CUPID)	Telix International Pty Ltd.	Phase 1	Australia
NCT04506567	Fractionated and Multiple Dose ^225^Ac-J591 for Progressive mCRPC	Weill Medical College of Cornell University	Phase 1/2	New York, USA
NCT05150236	EVOLUTION: ^177^Lu-PSMA Therapy Versus ^177^Lu-PSMA in Combination With Ipilimumab and Nivolumab for Men With mCRPC (ANZUP2001)	Australian and New Zealand Urogenital and Prostate Cancer Trials Group	Phase 2	Australia
NCT05413850	Anti-tumour Activity of (^177^Lu) rhPSMA-10.1 Injection	Blue Earth Therapeutics Ltd.	Phase 1/2	Maryland, USA
NCT04509557	[^177^Lu]Ludotadipep Treatment in Patients With Metastatic Castration-resistant Prostate Cancer.	FutureChem	Phase 1	Republic of Korea
** ^223^ ** **RaCl2**				
NCT04521361	A Study to Assess How Radium-223 Distributes in the Body of Patients With Prostate Cancer Which Spread to the Bones	Bayer	Phase 1	Multinational
NCT04037358	RAdium-223 and SABR Versus SABR for Oligometastatic Prostate Cancers (RAVENS)	Sidney Kimmel Comprehensive Cancer Center at Johns Hopkins	Phase 2	Maryland, USA
NCT03574571	A Study to Test Radium-223 With Docetaxel in Patients With Prostate Cancer	Memorial Sloan Kettering Cancer Center	Phase 3	Multinational
NCT05133440	A Study of Stereotactic Body Radiation Therapy and Radium (Ra-223) Dichloride in Prostate Cancer That Has Spread to the Bones	Memorial Sloan Kettering Cancer Center	Phase 2	USA
NCT03737370	Fractionated Docetaxel and Radium 223 in Metastatic Castration-Resistant Prostate Cancer	Tufts Medical Center	Phase 1	USA
NCT04109729	Study of Nivolumab in Combination w Radium-223 in Men w Metastatic Castration Resistant Prostate Cancer (Rad2Nivo)	University of Utah	Phase 1/2	Utah, USA
NCT04206319	Radium-223 in Biochemically Recurrent Prostate Cancer	National Cancer Institute (NCI)	Phase 2	Maryland, USA
NCT04597125	Investigation of Radium-223 Dichloride (Xofigo), a Treatment That Gives Off Radiation That Helps Kill Cancer Cells, Compared to a Treatment That Inactivates Hormones (New Antihormonal Therapy, NAH) in Patients With Prostate Cancer That Has Spread to the Bone Getting Worse on or After Earlier NAH	Bayer	Phase 4	Multinational
NCT03432949	Radium-223 Combined With Dexamethasone as First-line Therapy in Patients With M+CRPC (TRANCE)	Bayer	Phase 4	Canada
NCT04071236	Radiation Medication (Radium-223 Dichloride) Versus Radium-223 Dichloride Plus Radiation Enhancing Medication (M3814) Versus Radium-223 Dichloride Plus M3814 Plus Avelumab (a Type of Immunotherapy) for Advanced Prostate Cancer Not Responsive to Hormonal Therapy	National Cancer Institute (NCI)	Phase 1/2	USA
NCT04704505	Bipolar Androgen Therapy (BAT) and Radium-223 (RAD) in Metastatic Castration-resistant Prostate Cancer (mCRPC) (BAT-RAD)	Sidney Kimmel Comprehensive Cancer Center at Johns Hopkins	Phase 2	Multinational
NCT03361735	Radium Ra 223 Dichloride, Hormone Therapy and Stereotactic Body Radiation Therapy in Treating Patients With Metastatic Prostate Cancer	City of Hope Medical Center	Phase 2	California, USA
NCT02194842	Phase III Radium 223 mCRPC-PEACE III (PEACE III)	European Organisation for Research and Treatment of Cancer—EORTC	Phase 3	Multinational
NCT04704505	Bipolar Androgen Therapy (BAT) and Radium-223 (RAD) in Metastatic Castration-resistant Prostate Cancer (mCRPC) (BAT-RAD)	Sidney Kimmel Comprehensive Cancer Center at Johns Hopkins	Phase 2	Multinational
**GRPR**				
NCT05283330	Safety and Tolerability of ²¹²Pb-DOTAM-GRPR1 ²¹²Pb-DOTAM-GRPR1 in Adult Subjects with Recurrent or Metastatic GRPR- expressing Tumors	Orano Med LLC	Phase 1	Not yet recruiting

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
