# Peer review of "A Treatment Paradigm Shift: Targeted Radionuclide Therapies for Metastatic Castrate Resistant Prostate Cancer"

_cancers, 2022, doi:10.3390/cancers14174276_

Round 1
Reviewer 1 Report
Radiometabolic approaches combine specific prostate cancer cell ligands to radioactive particles, thus allowing to deliver cytotoxic radiations in cancer cells. Among these, radium-223 and lutetium-177 have shown promising activity in metastatic pretreated prostate cancer patients and further studies are ongoing to expand the applications of this therapeutic approach. In addition, nuclear medicine techniques also have an important diagnostic role in prostate cancer.
Thus, the study addresses a very timely, important topic in prostate cancer management.
Some changes are needed:
- We suggest to include a table summarizing the results of pivotal phase II/III trials of radiometabolic ligands in prostate cancer (ERA-223, ALSYMPCA, etc.)
- We recommend to add a table including all the ongoing clinical trials of radiometabolic ligands in prostate cancer (DORA, etc.)
- We suggest to expand the introduction section by adding further information regarding the medical treatment for prostate cancer patients and to include some recently published papers discussing this topic, only for a matter of consistency (PMID: 32911806 ; PMID: 33535541 )
Author Response
To the Editor and Reviewers of Cancers,
Thank you for your thoughtful evaluation of our manuscript. We appreciate your comments and have made changes as detailed below.
Reviewer 1:
- We suggest to include a table summarizing the results of pivotal phase II/III trials of radiometabolic ligands in prostate cancer (ERA-223, ALSYMPCA, etc.)
We have compiled a list of the Phase 2/3 studies that either led to the FDA approval of 223RaCl2/177Lu PSMA-617 or have been fundamental to help guide how these agents should be used in combination with hormonal therapy (Table 1). Published studies with α-PSMA, GRPR, or somatostatin therapeutic radiopharmaceuticals are predominately retrospective with no appropriate phase 2/3 clinical trials to include in Table 1.
- We recommend to add a table including all the ongoing clinical trials of radiometabolic ligands in prostate cancer (DORA, etc.)
We have compiled all ongoing clinical trails with targeted radionuclide therapies for prostate cancer as identified by an extensive search on clinicaltrials.gov on 08/26/2022 (Table 2). As far as we can ascertain, there are no ongoing clinical trials with α-PSMA, GRPR, or somatostatin therapeutic radiopharmaceuticals. We did include reference to a single upcoming GRPR trial. We note that this table is quite long and may be better served by inclusion as a supplemental file but we will leave that decision to the editor.
- We suggest to expand the introduction section by adding further information regarding the medical treatment for prostate cancer patients and to include some recently published papers discussing this topic, only for a matter of consistency (PMID: 32911806 ; PMID: 33535541)
We have done as the reviewer suggested and included a paragraph about current medical treatments for prostate cancer in the introduction, “Despite androgen independence, androgen deprivation therapy remains the backbone of treatment, in addition to, bone modifying agents and cancer-directed therapy. Metastatic disease to the bone poses great morbidity with skeletal-related events and pain, overall, negatively impacting quality of life. Bone modifying agents such as bisphosphonates (zoledronic acid) and receptor activator of nuclear factor κ B ligand (RANKL) inhibitor (denosumab) are necessary in CRPC patients with bone metastases to prevent SRE’s which are known to increase the risk of death and reduce quality of life [4,5] . There are multiple cancer directed therapeutic options available that improve overall survival (OS) in mCRPC which include androgen signaling inhibitors (abiraterone, enzalutamide), chemotherapy (docetaxel, cabazitaxel), autologous cellular immunotherapy (sipuleucel-T) and poly-ADP-ribose polymerase inhibitors (olaparib, rucaparib); however, despite these systemic therapies, mCRPC remains incurable[6].”
Reviewer 2 Report
Thank you for submitting a balanced overview of the advances with regards to targeted radionuclide therapies in prostate cancer with reference to the most important clinical trials and contributors in the field included.
Figures 1 and 2 might have to be reconsidered. The intended message to be portrayed with Fig 1 is not entirely clear and a bone scan is probably not the best imaging tool for follow-up evaluation.
Fig 2: the MIP is sufficient without the fused images. Consider replacing the Choline image with an 18F-PSMA-based image, as there is no reference to C-11-Choline in the text.
Consider including a short section on combination therapies.
Author Response
To the Editor and Reviewers of Cancers,
Thank you for your thoughtful evaluation of our manuscript. We appreciate your comments and have made changes as detailed below.
Reviewer 2:
Figures 1 and 2 might have to be reconsidered. The intended message to be portrayed with Fig 1 is not entirely clear and a bone scan is probably not the best imaging tool for follow-up evaluation.
While we share the concern that Figure 1 is not an ideal method to track the success of 223RaCl2, that is the point that we are making in the text. As 223RaCl2 is unable to be imaged itself, there is no good radiographic method to track therapy and delivery of radiation to known or suspected osseus metastatic disease. Standard of care MDP imaging is often employed and thus we used MDP to show how it is suboptimal for the nuclear radiologist to make firm conclusions as to the success of therapy. We have clarified this point in the text, “Assessing treatment response to 223RaCl2 with molecular imaging remains a challenge with commonly utilized bone specific radiopharmaceuticals (e.g. 99mTc methylene diphosphonate (MDP) as both benign healing and metastatic disease can have a similar presentation (Figure 1).”
Fig 2: the MIP is sufficient without the fused images. Consider replacing the Choline image with an 18F-PSMA-based image, as there is no reference to C-11-Choline in the text.
We have changed Figure 2 to include two different patients with mCRPC that underwent either Ga-68 PSMA-11 or F-18 PSMA DCFPyL PET in preparation of 177Lu PSMA-617 therapy. We have included MIP only images as suggested by the reviewer.
Consider including a short section on combination therapies.
We have discussed in detail in this review article the available results of combination hormonal therapy of 223RaCl2/177Lu PSMA-617. However, we feel that attempting to separate out the few available combination trail results to a dedicated paragraph will create a more convoluted discussion for the reader, especially considering the relative paucity of published clinical trials. Specifically, while there are a few studies documenting combination hormonal therapy with either 223RaCl2/177Lu PSMA-617, there are no published trials regarding combination chemotherapy/immunotherapy. Additionally, while there is an ongoing observational study in patients that received 177Lu PSMA-617 after 223RaCl2, only a small preliminary analysis of 26 patients has been published. We have discussed in detail how the practitioner should look at the available hormonal combination therapy data, but feel that trying to combine the paucity of data regarding combination 223RaCl2/177Lu PSMA-617 will not really benefit the reader. We have, as reviewer 1 suggested, included a table documenting the pivotal phase 2/3 studies (Table 1, inclusive of combination 23RaCl2 and hormonal therapy) and have also created a table of the current clinical trials with targeted radionuclide therapy as identified on clinicaltrials.gov, many of which concentrate on the concept of combination therapy.
Reviewer 3 Report
It is a well-written paper; the topics are structured properly and provide a quick overview on the latest available targeted radionuclide therapies for metastatic castration-resistant prostate cancer.
However, I have following recommendation:
- Please define the abbreviation SOC and correct the word “makers” to “markers” in line 99, page 3.
- It might be better to insert the captions of the images presented in the paper directly beneath the images.
Author Response
To the Editor and Reviewers of Cancers,
Thank you for your thoughtful evaluation of our manuscript. We appreciate your comments and have made changes as detailed below.
Reviewer 3:
- Please define the abbreviation SOC and correct the word “makers” to “markers” in line 99, page 3.
We have done as the reviewer suggested
- It might be better to insert the captions of the images presented in the paper directly beneath the images.
We have arranged the paper as indicated in the author instructions with a separate section for figures. We are happy to place the figures and captions in the text, but have left them in place awaiting editorial instructions.
Round 2
Reviewer 1 Report
acceptance.